# Graphene Film Growth on Silicon Carbide by Hot Filament Chemical Vapor Deposition

**DOI:** 10.3390/nano12173033

**Published:** 2022-09-01

**Authors:** Sandra Rodríguez-Villanueva, Frank Mendoza, Brad R. Weiner, Gerardo Morell

**Affiliations:** 1Department of Physics, College of Natural Science, Rio Piedras Campus, University of Puerto Rico, San Juan, PR 00925, USA; 2Molecular Sciences Research Center, University of Puerto Rico, San Juan, PR 00927, USA; 3Department of Physics, College of Arts and Sciences, Mayagüez Campus, University of Puerto Rico, Mayaguez, PR 00682, USA; 4Department of Chemistry, College of Natural Science, Rio Piedras Campus, University of Puerto Rico, San Juan, PR 00925, USA

**Keywords:** graphene, hot filament chemical vapor deposition, methane gas

## Abstract

The electrical properties of graphene on dielectric substrates, such as silicon carbide (SiC), have received much attention due to their interesting applications. This work presents a method to grow graphene on a 6H-SiC substrate at a pressure of 35 Torr by using the hot filament chemical vapor deposition (HFCVD) technique. The graphene deposition was conducted in an atmosphere of methane and hydrogen at a temperature of 950 °C. The graphene films were analyzed using Raman spectroscopy, scanning electron microscopy, atomic force microscopy, energy dispersive X-ray, and X-ray photoelectron spectroscopy. Raman mapping and AFM measurements indicated that few-layer and multilayer graphene were deposited from the external carbon source depending on the growth parameter conditions. The compositional analysis confirmed the presence of graphene deposition on SiC substrates and the absence of any metal involved in the growth process.

## 1. Introduction

In the search to discover new technologies, such as those that emerged during the silicon (Si) era, considerable research has been focused on exploring similar materials that contribute to electronic innovations. Graphene, a two-dimensional structure, looks like a promising material for the next generation of electronic devices, due to its favorable properties, e.g., high electron mobility of 200,000 cm^2^ V^−1^ s^−1^ and a carrier concentration of 10^12^ cm^−^^2^ [1]. There are many techniques to obtain graphene, including microexfoliation [2], graphene oxide reduction [3], epitaxial growth on SiC [4,5], and chemical vapor deposition (CVD) [4,6,7]. This CVD methodology is the most widely used and efficient method to grow graphene [8,9] and can be further subdivided into thermal chemical vapor deposition (TCVD) [10,11], hot filament chemical vapor deposition (HFCVD) [12,13,14], and plasma-enhanced CVD (PECVD) [15,16]. Among these, the HFCVD method has been shown to be a systematic and easier way to control the growth parameters. This technique allows two thermal points (holder temperature and filament) to be set independently, resulting in a temperature gradient on the sample [12,13,14]. To take advantage of its excellent electrical properties, graphene must be deposited on particular substrates, i.e., dielectrics, such as silicon carbide (SiC). Therefore, many studies have focused on finding the best method to grow graphene on semiconducting materials [17,18,19]. Specifically, SiC appears to be a remarkable material for electronic applications due to its wide bandgap, high thermal conductivity, excellent thermal/chemical stability, and other properties that make this material a good candidate for high-temperature and high-power applications [20,21,22]. There are ~170 polytypes of SiC with different interesting properties formed under ambient conditions [20,23]. According to the stacking sequence, SiC can form three different structures: cubic, hexagonal, and rhombohedral [23]. However, the most prevalent types correspond to hexagonal (4H, 6H, or α) and cubic (3C or β) structures. The combination of these two exceptional materials, graphene and SiC, has attracted a lot of interest because they mutually improve their shared electrical properties, such as the reduction in sheet resistance in this system compared to normal SiC [24,25]. The first approach to grow graphene on SiC was carried out by Badami in 1965 on a hexagonal substrate at 2180 °C in a vacuum environment [26]. In 1975, Bommel et al. were able to form a graphite monolayer on hexagonal SiC for the Si and C faces at 800 °C in ultra-high vacuum (UHV) [27]. Based on these experiments, researchers used this thermal decomposition method to obtain graphene on SiC [25,28,29,30,31,32] under different conditions. In 2004, Berger et al. recorded the first transport measurements of multilayer graphene on SiC obtained in UHV [33] and found that the electron mobility was ca. 1100 cm^2^ V^−1^ s^−1^. Since then, many researchers have focused on improving the quality of graphene on SiC. For example, Rollings et al. obtained graphene at 1200 °C in UHV [31], while Boeckl’s group followed up their experiments in a vacuum environment, but at a higher temperature [34]. However, the high sublimation rate of Si in thermal decomposition experiments impairs the quality of graphene [4], which led to the exploration of methodologies involving gas flows, e.g., argon, to maintain the excellent properties of this carbon material. Emtsev et al. grew graphene on 6H-SiC in an argon environment (900 mbar) at 1650 °C [35], and their results confirmed an improvement in the graphene’s quality according to Raman and Hall Effect measurements. Similar experiments were conducted in hydrogen [36] and Si [37] environments to optimize the films’ properties. New methodologies, such as the application of a confinement-controlled sublimation system to reduce the Si evaporation rate [38] and nickel film deposition on SiC to induce graphitization at lower temperatures [24], were implemented.

Several studies have shown that reducing the sublimation rate of Si improves the quality of graphene [39,40,41,42]. Consequently, some works have proposed the incorporation of the external carbon source, such as propane and methane [39,40,41,42], to reduce the Si sublimation effect. In one study conducted by Dagher et al., graphene growth experiments were carried out in a different mixture of gases (Ar, H_2_, and CH_4_), temperatures, and pressures [39]. They concluded that the effect of Si sublimation was reduced due to the presence of H_2_, thus inhibiting the carbon layer formation. Consequently, graphene growth on SiC was only possible by including the CH_4_ gas as external carbon source [39]. Similarly, Liu et al. proposed a graphene growth method using methane as a carbon source. Their experiments were divided into three steps: hydrogen etching, nucleation, and growth [41]. In the first step, uniform flat terraces were obtained on the substrate, and in the second step using an argon atmosphere, carbon nuclei were formed from the SiC substrate. Finally, the graphene growth was completed with the flow of methane [41]. These results demonstrated that the use of the CVD method with an external carbon source provides a clear advantage, i.e., the reduction in the Si sublimation effect, avoiding the influence of the substrate, which could incorporate defects into the graphene film. In contrast to other methods relying only on the Si sublimation process, the CVD technique with an external carbon source has delivered the highest quality graphene. We present here a method to grow high-quality graphene on 6H/SiC, C-terminated face by HFCVD, using methane as the carbon source. The experiments were conducted at a temperature of 950 °C and a pressure of 35 Torr. The graphene films were characterized by Raman, SEM, AFM, EDS, and XPS measurements.

## 2. Materials and Methods

### 2.1. Substrate Preparation

The silicon carbide substrates (1 cm × 1 cm) correspond to the C-terminated α-SiC (6H) hexagonal structures from Valley Design Corp. (Shirley, MA, USA, http://valleydesign.com/, accessed on 25 July 2022) with a thickness of 508 ± 50 µm. The substrates were cleaned sequentially with trichlorethylene (TCE), acetone, deionized water, and isopropanol, followed by a H_2_SO_4_:H_2_O_2_ mixture to eliminate organic residues. In some cases, a 50% hydrofluoric acid (HF) treatment was used as extra cleaning to remove the oxide on the SiC surface [40] and to analyze the effect on the graphene quality. Two groups of substrates were prepared, one with 10 min of HF treatment and one without HF acid treatment. Other HF treatment exposure times (from 0 to 360 min) were tried, but no difference was found. All reagents listed above were obtained from Fisher Scientific (Pittsburgh, PA, USA; https://www.fishersci.com/, accessed on 25 July 2022).

### 2.2. Graphene Synthesis

The SiC substrates were introduced into the HFCVD equipment (BWS-HFCVD1000, Blue Wave, Baltimore, MD, USA; https://www.bluewavesemi.com/, accessed on 25 July 2022) [14], and the chamber was evacuated to 1 × 10^−3^ Torr. The graphene deposition procedure was then carried out in two steps: (1) annealing and (2) growth. In the anneal-ing process, the SiC substrates were heated at 950 °C and exposed to a mixture of 80 sccm of hydrogen and 20 sccm of argon for 30 min. For the growth process, the argon gas flow was stopped, and filaments were turned on, over a temperature range of 1800 to 2300 °C. Methane gas (1–10 sccm) and H_2_ gas (0–50 sccm) were flowed, and the graphene growth process was performed for different time periods (30–300 min) and different experimental flow conditions. Following the growth process, the filament and heater were turned off, and the sample was allowed to cool to ambient conditions (see Figure 1b). All experiments leading to graphene deposition (annealing and growth) were conducted at a constant pressure of 35 Torr.

### 2.3. Characterization

The presence and characteristics of graphene on SiC were verified and analyzed using multiple techniques. Raman spectroscopy was performed by a micro-Raman system (Thermo Scientific DXR, Waltham, MA, USA) with a 532 nm laser as excitation source, and all measurements were taken at atmospheric pressure and ambient temperature. Raman scattering spectra were collected over a spectral range of 1100–3100 cm^−1^, using a focused spot size of 0.7 μm. Raman measurements made it possible to estimate the characteristics of graphene, such as the number of graphene layers and the size of the crystal. Raman mappings were obtained in areas of 150 × 110 μm^2^ with a step size of 2 μm; the collection time for each point was 20 s. A scanning electron microscope, SEM (IT500HR, JEOL, Peabody, MA, USA; https://www.jeolusa.com/, accessed on 29 August 2022) and an atomic force microscope, AFM (Nanoscope V, Vecco, Plainview, NY, USA; https://www.veeco.com/, accessed on 25 July 2022) were used to study the morphological characteristics of graphene crystals, as well as their size and distribution. The SEM and AFM images were processed by SMILE VIEW Lab software (from JEOL and Digital Surf, Besançon, France) and by Nanoscope 8.15 software (Bruker corporation, Santa Barbara, CA, USA), respectively. AFM measurements were taken at atmospheric pressure, and the SEM was carried out at a pressure of 1 × 10^−8^ Torr. The compositional analysis was performed using energy-dispersive X-ray spectroscopy, EDS (DRYSD30, JEOL, Chanhassen, MN, USA; https://www.phi.com/index.html, accessed on 25 July 2022), and X-ray photoelectron spectroscopy, XPS (PHI 5600 Physical Electronics, Chanhassen, MN, USA), with a voltage of 20 kV and an energy range of 0 to 1200 eV, respectively. The EDS measurements were conducted at the same pressure as the SEM, and the XPS data were taken at 1 × 10^−10^ Torr. The XPS data were analyzed after the deconvolution of C1s peaks using the Origin program with the Gaussian function fitting and the EDS spectra by SMILE VIEW Lab software (from JEOL and Digital Surf, Besançon, France). The XPS and EDS analyses have an average depth of approximately 5 nm and 2 µm, respectively.

## 3. Results

Graphene growth on 6H-SiC samples under different conditions was characterized using the Raman technique to estimate the number of deposited layers and defects in the films. SEM and AFM allowed a deeper study about the morphology, and EDS and XPS were used to identify the elements present in the graphene samples. 

### 3.1. Raman Analysis

The graphene growth on SiC substrates (treated with and without HF acid) was characterized by the D, G, 2D, and SiC peaks of the Raman spectra. Figure 2a,b show the signal obtained for samples grown at 10 sccm and 1 sccm of methane flow, respectively, that were previously cleaned with HF. In addition, Figure 2c,d show the same but for samples prepared with untreated (no HF) substrates. The growth conditions of all experiments are summarized in Table 1. The D, G, and 2D peaks were observed at ~1349 cm^−1^, ~1580 cm^−1^, and at ~2694 cm^−1^, respectively (cf. Figure 2). SiC peaks were observed at 1526 cm^−1^ and 1715 cm^−1^, showing that few layers of graphene were grown and that the Raman signal of the substrate is not eliminated by carbon deposition [13,34].

However, in experiments with higher methane fluxes or longer growth times, the intensity of SiC peaks was reduced. Nevertheless, multilayer graphene was obtained on substrates that were previously cleaned with HF and in experiments at longer growth time and with a flow of 1 sccm (cf. Table 1). The characteristics of the D and G peaks provide information about the graphene quality [13,43,44,45] and the crystal size [46,47,48], where the D peak is related to the defect and disorder of the sp^2^ carbon network [13,24,43,44]. The D/G intensity ratio was used to obtain information about the defect level in the films, where high values were associated with a more defective structure [24]. Our results show that the D/G intensity ratio for a few layers (cleaned and uncleaned substrates) was between 0.20 ± 0.03 and 0.80 ± 0.03, which indicates a low defective crystal structure [24], while in multilayer samples (cleaned substrates), this value was between 1.70 ± 0.02 and 1.90 ± 0.03. 

We established that the presence of this peak (D) could be related to the nanometric particle size of the carbon crystals [49] and to the strong scattering caused by defects in the crystal structure [13,50,51,52]. In addition, the full width at half maximum (FWHM) values of the graphene peaks gives us more information about the crystallinity and quality of this carbon material [13,50]. The results show that the D, G, and 2D FWHM values correspond to 30 cm^−1^, 30 cm^−1^, and 60 cm^−1^ for few-layer films grown on cleaned and uncleaned surfaces with HF and 48 cm^−1^, 60 cm^−1^, and 90 cm^−1^ in multilayer samples prepared on cleaned surfaces.

Although we obtained the FWHM of the 2D peak in the range of 60 to 90 cm^−1^, indicating low crystallinity, these results were compared with the single-layer (60 cm^−1^) and bilayer (90 cm^−1^) epitaxial graphene obtained in Ni’s work [29]. The D/G intensity ratio was used for the estimation of the particle size of graphene [44,47]. Initially, Tuinstra and Koening proposed a relation using this ratio (D/G) for Raman spectra excited with a radiation source of 514.5 nm [44]. Cancado et al. later established a relation (Equation (1)) between excitation wavelength (*λ_l_*) [47] and particle size, where *Lα* represents the particle size, the value 2.4 × 10^−10^ is a proportionality constant, and *I_D_/I_G_* is the D/G intensity ratio:(1)$$L_{\alpha }=\left(2.4\times 10^{-10}\right)\;\lambda_{l}^{4}\;{\left(\frac{I_{D}}{I_{G}}\right)}^{-1}$$

According to Cancado’s relation, the nanometric crystallites were around 23.50 nm to 90.00 nm for few layers samples and 8 nm to 12 nm in multilayer films. Although these results were close to the particle size estimated by SEM in the few layers (30 to 100 nm) and multilayer samples (10 nm), some discrepancies were observed. To understand these discrepancies, Equation (2) was employed to calculate the crystal’s defect, where E_L_ corresponds to the excitation energy, *L_D_* is the inter-defect distance, and 1/*L_D_* is the defect concentration [53]. These calculations resulted in *L_D_* values of 24 nm and 8 nm for few layer and multilayer films, respectively, and defect concentrations of 1.30 × 10^−3^/nm^2^ and 1.55 × 10^−2^/nm^2^, correspondingly:(2)$$L_{D}^{2}\;\left({\mathrm{nm}}^{2}\right)=\frac{3600}{E_{L}^{4}}{\left(I_{D}/I_{G}\right)}^{-1}$$

### 3.2. Raman Mapping

To obtain information about the number of graphene layers and the uniformity of these films on SiC, a Raman map was obtained in areas of 150 × 110 µm^2^. These measurements were derived from the intensity ratio of the 2D/G peaks [13,47]. Figure 3a shows the image of the selected mapping area of the 2D/G intensity ratio for a few layers (cleaned and uncleaned substrates) of graphene, where the blue and red colours correspond to the lowest and highest values, respectively. These values were in the range of 0.5 ± 0.03 and 0.6 ± 0.03, confirming that a few layers (6 to 12 layers) were deposited [13,46,50,54]. Similarly, Figure 3b shows the image corresponding to the intensity ratio of the 2D and G peaks of the mapped area in the multilayer (cleaned substrates) graphene samples. In this case, the values (2D/G) were in the range of 0.3 ± 0.05 to 0.4 ± 0.05, but with less uniformity than in the few layer samples. While the 2D/G intensity ratio provides an estimate of the number of graphene layers [13,17,29], other aspects, e.g., the level of doping in graphene films, should also be considered because they can influence this ratio (2D/G) as well [55].

According to our results, the best graphene quality and uniformity were achieved using two methods: (1) the substrates (without HF treatment) were exposed to a methane flux of 1 sccm for 300 min at 950 °C, and (2) the SiC substrates were cleaned with HF and then heated at 950 °C in an atmosphere of 10 sccm of methane for 60 or 120 min. A significant observation was that graphene grew on cleaned and uncleaned SiC substrates. Nevertheless, the results demonstrated that more graphene layers were deposited on substrates cleaned with HF than in the untreated substrates. This outcome is consistent with the HF removing the oxygen on the surface and allowing for better carbon incorporation [40]. However, Dhar et al. reported that an oxygen monolayer with an OH termination was still present on the SiC surface after the exposition to HF acid [56]. Therefore, it is possible to find residual oxygen after the HF treatment, which will have an effect in the growth of graphene. For this reason, we always performed an annealing process. In summary, it was possible to grow few-layer graphene on both treated and untreated SiC substrates, but Raman spectra show that multilayer graphene was obtained only on HF cleaned substrates for longer growth times. SEM, AFM, EDS, and XPS characterization show that the morphology and composition of the graphene layers were indistinguishable between clean and uncleaned substrates. For this reason, the following analysis will be limited to the graphene grown on SiC substrates cleaned with HF.

### 3.3. SEM Analysis

The SEM images of graphene on SiC for few-layer and multilayer samples, respectively, at 140,000× magnification, are shown in Figure 4a,b. Figure 4c,d show the images of the same samples but at 25,000× magnification. We observed, in both images, the nanometer scale particles over all samples with an average grain size of 30 to 100 nm in few-layer samples. However, in the multilayer films, fibre-like particles, 10 nm in size, were evenly distributed throughout the SiC surface. The graphene particle size obtained by SEM, are in reasonable agreement with the estimates calculated by the Cancado equation [47] (D/G peaks) above.

### 3.4. AFM Analysis

The morphology of graphene on SiC was further characterized by AFM measurements taken in an area of 3 µm × 3 µm. Figure 5a shows the nanometer-scale crystal with a height of 20 nm and a diameter of around 100 nm, corresponding to few layer graphene [35]. The AFM results of the graphene morphology are consistent with the SEM images, confirming that the graphene nanocrystals were grown on SiC. Figure 5b corresponds to the AFM image of the SiC substrate, where it was possible to observe some lines and scratches from the manufacturing process. Further analysis by AFM of the SiC substrate revealed that after the HF cleaning treatment and the annealing process, the surfaces did not improve (see Appendix A).

### 3.5. EDS Analysis

As expected, elemental carbon and silicon were identified as the major components on the EDS spectra. The few-layer graphene samples show an atomic concentration of 60.41 ± 3.00% of carbon, 37.73 ± 3.00% of silicon, and 1.86 ± 3.00% of oxygen (Figure 6a). For multilayer samples, the percentage was 69.47 ± 3.00%, 29.90 ± 3.00%, and 0.63 ± 3.00% to carbon, silicon, and oxygen, respectively (Figure 6b). By comparison, SiC substrates showed a percentage of 49.09 ± 3.00%, 50.68 ± 3.00%, and 0.32 ± 3.00% for carbon, silicon, and oxygen, respectively. No other elements were found in the samples, e.g., copper, nickel, or rhenium (filament) that could modify the mechanism of the growth reactions.

### 3.6. XPS Analysis

XPS measurements were performed before and after graphene deposition on the SiC substrate. Figure 7a,b show the full spectrum and carbon peak of graphene growth on SiC, respectively. Similarly, the XPS spectra of the bare SiC substrate and C peak are shown in Figure 7c,d. As expected for both samples (before and after graphene growth), elemental carbon, silicon, and oxygen were present in the spectra. No trace metals were found on the surface, consistent with the EDS measurements. After deconvolution of the C1s peak of graphene on SiC samples, two peaks, 283 and 285 eV corresponding to SiC and graphene, can be resolved [24,41,57]. These peaks correspond to the characteristic signal of graphene grown on the carbon-terminated face of SiC [58]. The deconvolution of the C1s peak of the bare SiC substrate showed a peak at 283 eV corresponding to the SiC bond. A second peak (X) at 285.7 eV and a third peak (Y) at 287.6 eV were found, which are generally associated with C–O and C=O bonding on the surface [41,59,60], respectively (Figure 7d)_._ Oxygen incorporation can occur during sample preparation for XPS testing. In addition, we observed a reduced intensity in the SiC peak after the growth of graphene [30] (see Figure 7b).

## 4. Growth Mechanism

In the present work, nanographene films were grown on SiC substrate by HFCVD using methane as carbon source. We propose that the graphene growth mechanism occurs in two steps: (a) the dehydrogenation of methane by the hot filaments produces reactive carbon species, CHx and C (Figure 8a), and (b) these reactive species deposit carbon atoms, stabilizing the SiC/C surface and forming the graphene film [34,61] (Figure 8b). Annealing prior to the growth process was conducted to clean the surface from contaminants [28] and oxygen [62,63], as well as to promote a flat terrace on SiC [64,65]. We discarded the graphene growth by Si sublimation [25] during the annealing, given that the gas mixing (Ar/H_2_), the pressure (35 torr), and temperature (950 °C) were not the correct conditions for this process [39]. This was confirmed by Raman measurements, where the graphene signal was not observed for SiC substrates with just the annealing. For this reason, the graphene growth experiments on SiC at the above-mentioned conditions were conducted, with methane as an external carbon source. The addition of H_2_ works as a carrier gas and reduces the formation of defects as it promotes better uniformity in the graphene film [42]. The SiC substrates cleaned with HF showed more graphene layers at the same growth conditions. The HF helps to remove the oxide layer at the SiC surface, allowing for better carbon incorporation at the graphene growth step. The growth time also has an effect on the number of deposited layers, i.e., longer growth time results in more graphene layers. In summary, the deposition of graphene on SiC is possible independently of the HF treatment. Nevertheless, the type of treatment and the growth time has an influence in the number of graphene layers. These results open the possibility of obtaining graphene on SiC using HFCVD at relatively low temperature and high pressure (not UHV) compared to other methods, which allows easier incorporation for mass production.

## 5. Conclusions

Graphene growth on 6H-SiC was performed by HFCVD at 950 °C and high pressure (not UHV) using methane as the external carbon source. Raman, SEM, AFM, EDS, and XPS measurements confirmed that a few layers of graphene were deposited at nanometer size. The surface morphology analysis indicated that high-quality graphene was obtained, and further cleaning with HF acid increased the number of carbon layers. A comparative analysis of the AFM measurements of SiC substrates with and without HF treatment showed no differences in surface quality. This work establishes a method for graphene growth on SiC with controllable parameters under different conditions to adjust the characteristics of graphene according to the application. Furthermore, these experiments open the possibility for graphene production by HFCVD method, which is suited for scaling in industrial applications.

## Figures and Tables

**Figure 1 nanomaterials-12-03033-f001:**
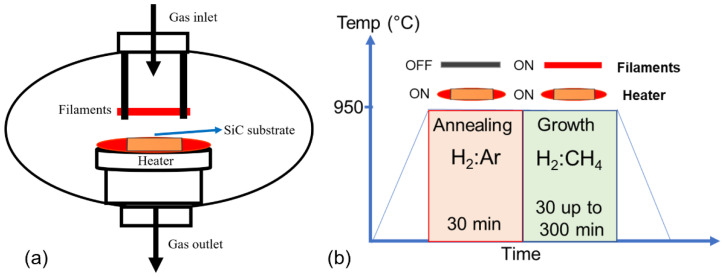
Hot filament chemical vapor deposition (HFCVD) reactor and graphene growth process, (**a**) schematic and (**b**) graphene growth steps on the SiC.

**Figure 2 nanomaterials-12-03033-f002:**
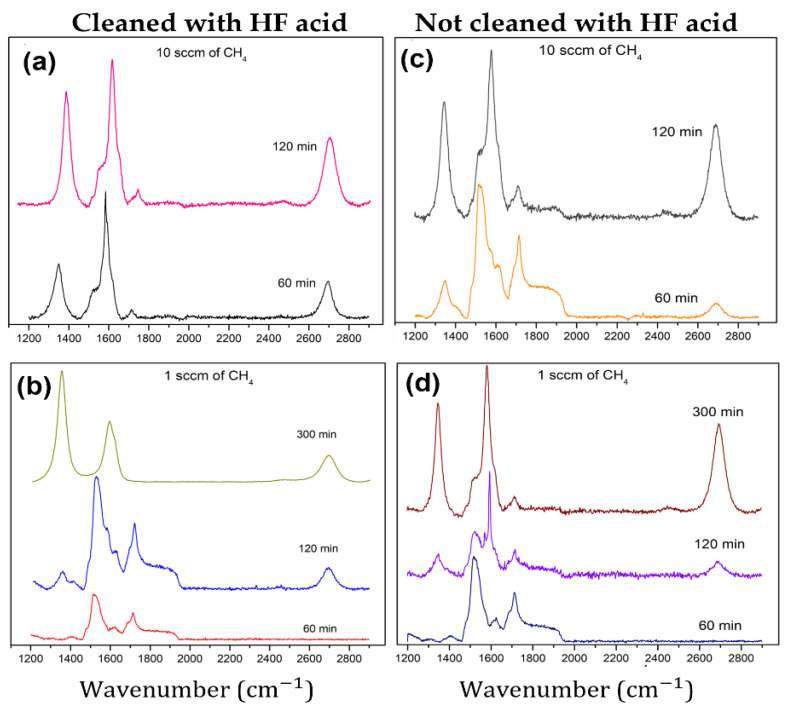
Raman spectra of graphene on SiC substrate: (**a**,**b**) represent the samples cleaned with HF and exposed to 10 and 1 sccm of CH_4_ flow, respectively, and (**c**,**d**) correspond to samples at the same gas flow but without the HF cleaning process, respectively.

**Figure 3 nanomaterials-12-03033-f003:**
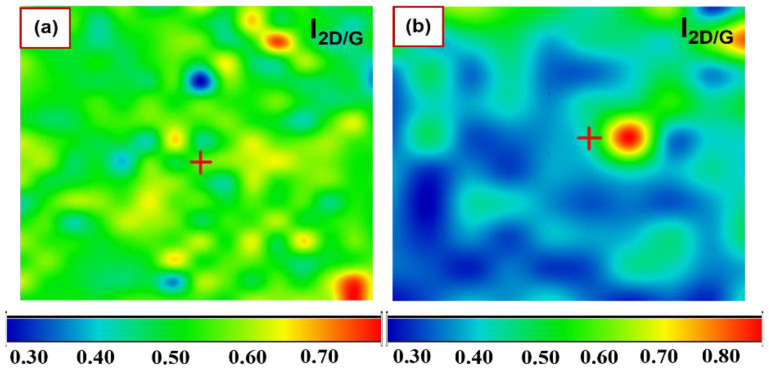
Raman mapping of graphene on SiC substrate: (**a**,**b**) represent the mapping image of the 2D/G intensity ratio for few-layer and multilayer samples, respectively.

**Figure 4 nanomaterials-12-03033-f004:**
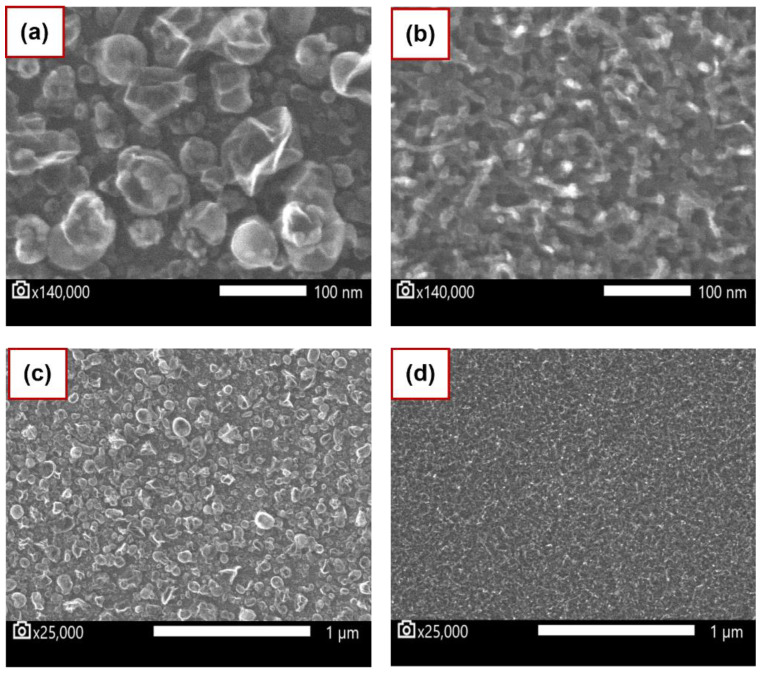
SEM measurements of the graphene growth on SiC substrate: (**a**,**b**) show the SEM image taken in few layers and multilayers samples at 140,000× magnification. Similarly, (**c**,**d**) show the same but at 25,000× magnification.

**Figure 5 nanomaterials-12-03033-f005:**
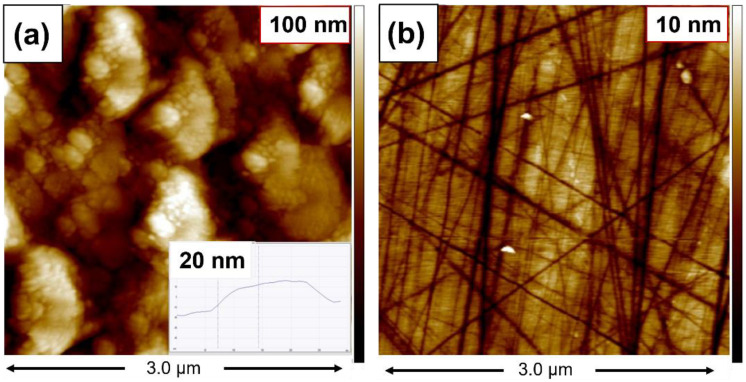
AFM measurements: (**a**,**b**) show the AFM images taken in graphene growth on SiC and on the SiC substrate, respectively.

**Figure 6 nanomaterials-12-03033-f006:**
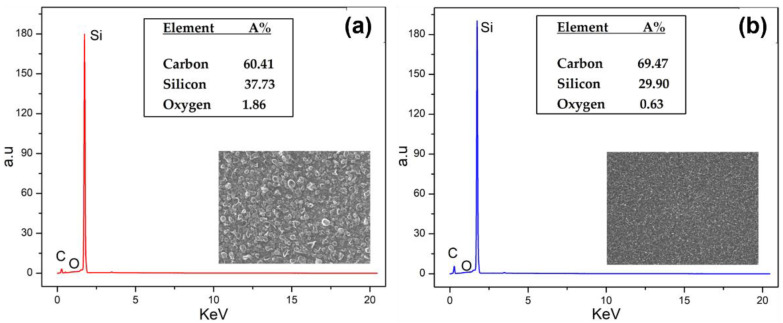
EDS spectrum of graphene on SiC in few layers (**a**) and multilayer (**b**) samples, respectively.

**Figure 7 nanomaterials-12-03033-f007:**
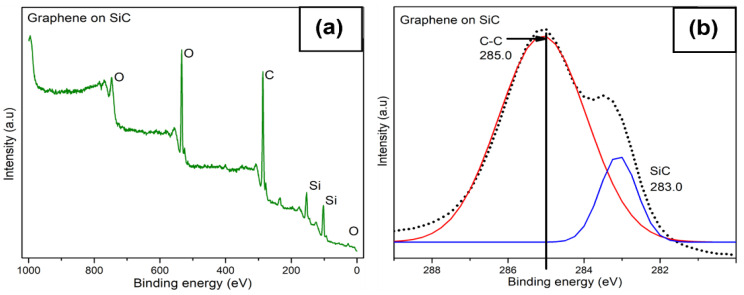
XPS measurements of full composition spectra (**a**) and carbon peak (**b**) after deconvolution of graphene on SiC, respectively; (**c**,**d**) correspond to the same but for the bare SiC substrate prior to deposition of graphene.

**Figure 8 nanomaterials-12-03033-f008:**
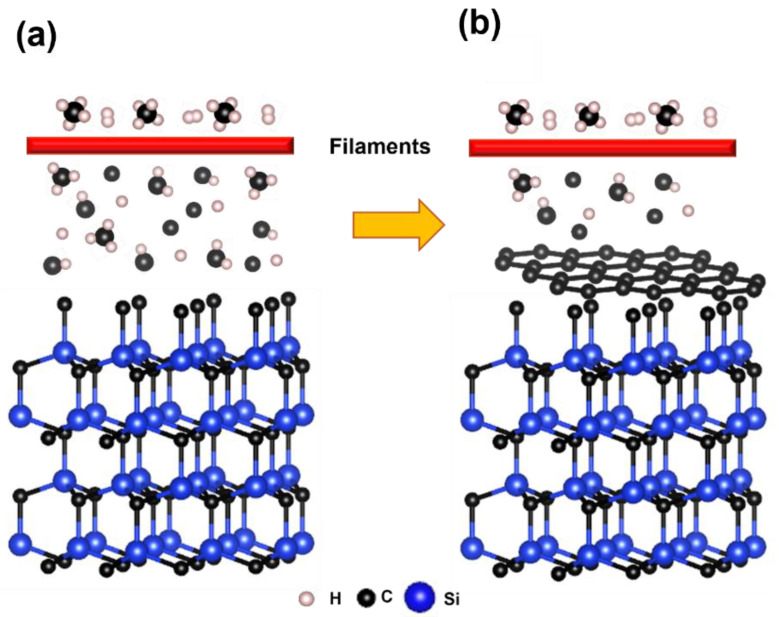
Graphene growth mechanism on SiC. (**a**) the dehydrogenation of methane by the hot filaments and (**b**) the reactive species deposit carbon atoms, stabilizing the SiC/C surface to form the graphene film.

**Table 1 nanomaterials-12-03033-t001:** Graphene on SiC—Growth conditions and Raman analysis.

Condition	Growth Parameters				Graphene Films Characteristics
	CH_4_ (sccm)	H_2_ (sccm)	Time (min)	Temperature (°C)	
Cleaned with HF	10	50	120	950	Few layers
	10	50	60	950	Few layers
	1	-	300	950	Multilayers
	1	-	120	950	Few layers
	1	-	60	950	No growth
Not cleaned with HF	10	50	120	950	Few layers
	10	50	60	950	Few layers
	1	-	300	950	Few layers
	1	-	120	950	Few layers
	1	-	60	950	No growth

## Data Availability

All data can be obtained from the corresponding author.

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
