# Peer review of "Graphene Film Growth on Silicon Carbide by Hot Filament Chemical Vapor Deposition"

_nanomaterials, 2022, doi:10.3390/nano12173033_

Round 1

Author Response

  1. Reviewer #2:

COMMENTS:

This manuscript presents a method to grow graphene on 6H-SiC substrate at low pressure (35 Torr) by using the hot filament chemical vapor deposition (HFCVD) technique. The manuscript can be accepted after minor revision. Authors should make the following corrections:

We appreciate the suggestions to improve the presentation of this work.

CORRECTIONS:

  1. More details to the section “Materials and Methods” should be added, especially about XPS measurements and fitting procedure for X-ray photoelectron spectra.

Additional information about the Materials and methods was included in pages 3 and 4.

  1. In the section 3 “Results”: In the part with discussion of Raman spectra the references to up-to date investigations in this field [J. Alloys Compd. 2020, 849, 156699] should be added

The reference was included in the manuscript.

  1. For fitting od C1s with BE~285 eV should be used asymmetric line and the references to up- to date investigations in this field [2D Materials, 2020, 7(4), 045001] should be added.

Figures 7 b, d was modified. The reference was included in the manuscript.

  1. XPS fitting: how about component for sp3 carbon?

According our XPS measurements the component related to sp3 and sp2 are not cleared differentiated after deconvolution process, because the resolution of our equipment. We have a peak at 284.8 eV that could be a combination of these components (sp2 and sp3)

Gratefully,

Sandra Rodriguez Villanueva

Reviewer 2 Report

Article „Graphene Film Growth on Silicon Carbide by Hot Filament 2 Chemical Vapor Deposition” describes a method to grow graphene on 6H-SiC substrate at low pressure by using the hot filament chemical vapor deposition (HFCVD) technique and characterizes the results of this technique with various analytical methods. The the article is written carefully with a small number of understatements. After the corrections article should be worth publishing.

I do not see the necessity of presenting Figure 3. After what is written, this Figure provides nothing new. Could the authors at least estimate a percentage of the area with the highest D/G ratio to claim that statistically, Fig a has, e.g. 10% area with defects and b) 5%?

Using such a small error, “37.73 ± 0.04% of silicon” with the EDS  method is unjustified. I understand that the equipment spits it out, and it is easy to copy that way, but the measurement will be different in different places, changing the value by a few percentage points. The for sure is higher. Please use the average error.

Text corrections:

Line 33 upper index “1012”

Line 95 “The experiments were conducted at low temperature (950 °C), and high pressure (35 Torr) compared to other methods.” Using low and high is confusing without writing in relation to what it is low or high. The best way is to omit words low and high because without context; they do not mean anything.

Line 119 For clarity, was the graphene growth process conducted without Ar flow in pure methane and hydrogen? Please check that.

Line 158  The laser-focused light has a micrometre length in the vertical direction. You should be able to see the substrate unless the surface screens the light well. XPS is better for surface studies on this scale than Ramman spectroscopy.

Line 212 can you also add the number of layers?

Author Response

Reviewer #2:

COMMENTS:

Article Graphene Film Growth on Silicon Carbide by Hot Filament 2 Chemical Vapor Deposition” describes a method to grow graphene on 6H-SiC substrate at low pressure by using the hot filament chemical vapor deposition (HFCVD) technique and characterizes the results of this technique with various analytical methods. The the article is written carefully with a small number of understatements. After the corrections article should be worth publishing.

We appreciate your recommendations.

CORRECTIONS:

  1. I do not see the necessity of presenting Figure 3. After what is written, this Figure provides nothing new. Could the authors at least estimate a percentage of the area with the highest D/G ratio to claim that statistically, Fig a has, e.g. 10% area with defects and b) 5%?

Figure 3 corresponds to the Raman mapping of graphene on SiC substrate, where a) and b) represent the mapping image of the 2D/G intensity ratio for few layers and multilayers samples, respectively. The reason to present this figure is to show the uniformity and distribution of the number of graphene layers in two types of samples: few and multilayers.

  1. Using such a small error, “37.73 ± 0.04% of silicon” with the EDS method is unjustified. I understand that the equipment spits it out, and it is easy to copy that way, but the measurement will be different in different places, changing the value by a few percentage points. The for sure is higher. Please use the average error.

We agree, the error percentage was changed. I contacted the JEOL company, they suggested an average error of 3.00%.

  1. Line 33 upper index “1012”

We corrected the carrier concentration value.

  1. Line 95 “The experiments were conducted at low temperature (950 °C), and high pressure (35 Torr) compared to other methods.” Using low and high is confusing without writing in relation to what it is low or high. The best way is to omit words low and high because without context; they do not mean anything.

We corrected the sentence.

  1. Line 119 For clarity, was the graphene growth process conducted without Ar flow in pure methane and hydrogen? Please check that.

Yes, the graphene growth process was conducted with pure methane and hydrogen. Some experiments were carried out without hydrogen. In the table 1, all growth parameters are described.

  1. Line 158 The laser-focused light has a micrometre length in the vertical direction. You should be able to see the substrate unless the surface screens the light well. XPS is better for surface studies on this scale than Ramman spectroscopy.

We add this sentence (“SiC peaks were observed at 1526 cm-1 and 1715 cm-1, showing that few layers of graphene were grown, and that the Raman signal of the substrate is not eliminated by carbon deposition”) because in multilayers samples the SiC peaks were not observed.

  1. Line 212 can you also add the number of layers?

We added the number of few layers on line 231.

Round 2

Reviewer 2 Report

The authors corrected the text as advised. I see no more problems with this text.